


# Fortnight conditioning of historical data to improve short-term precipitation predictions

Yoshito Hirata[1] and Yoshinori Yamada[2]

[1]Faculty of Engineering, Information and Systems, University of Tsukuba, 1-1-1 Tennodai, Tsukuba, Ibaraki 305-8573, Japan
[2]Eikei University of Hiroshima, 1-5 Nobori-cho, Naka-ku, Hiroshima-shi, Hiroshima 730-0016, Japan

**Correspondence:** Yoshito Hirata (hirata@cs.tsukuba.ac.jp)

**Abstract.** The effects of changes in weather variables, including precipitation dependence on the days-of-the-week, have known applications in weather predictions. However, the use of these effects to improve weather forecasting has not been determined. Here we investigate if conditioning past data somehow by considering the days-of-the-week helps us to obtain the better short-term time series prediction for precipitation. Especially, we demonstrate that short-term time series prediction of precipitation up to 2 h ahead can be improved using the data points of the days whose differences from the current day are multiples of 14. For short-term predictions, we employ infinite-dimensional delay coordinates (Hirata et al., Sci. Rep. 5, 15736, 2015) to reconstruct the underlying dynamics. Although the results demonstrate that the two-week periodicity seems to exist in the weather at Tokyo, and thus some anthropogenic activities could influence weather, the mechanism of the influence remains unclear.

## 1 Introduction

Human activities are considered as an origin of a weekly cycle because there is no natural cause of such cycle (Sanchez-Lorenzo et al. 2012). There are two main paths that human activities influence the weather in a weekly manner: The first path is aerosols and the second path is heats (Simmonds and Keay 1997; Sanchez-Lorenzo et al. 2012).

Due to transportation and industrial plants, more aerosols are generated on weekdays than on weekends (Rosenfeld 2014). Aerosols and related convective energy are related in a uni-modal fashion (Rosenfeld et al. 2008) because aerosols both scatter and absorb solar radiation (Tao et al. 2012). Therefore, anthropogenic aerosols are considered to lead to more thunderstorms in the middle of weeks in the summer in south-eastern United States (Bell et al. 2008; Rosenfeld 2014). It is also observed that more heats are generated in urban area on weekdays than weekends (Fujibe 2010), typically observed in Tokyo and Osaka.

Regardless of the existing controversy regarding whether weekly cycles are present in meteorological variables, we have devised a scheme that considers such weekly cycles to enhance the accuracy of short-range precipitation forecasts. This paper describes its mathematical scheme based on time series forecasting up to 2 hours ahead. Its performance will then be demonstrated by the application of the method to observed data at Tokyo. The discussions on why the incorporation of such weekly cycles improves the short-range forecasts is out of scope of this paper.





## 2 Methods

In this study, the weather variable of precipitation amount measured every 1 min was used. We used the datasets generated by the Japan Meteorological Agency, which were observed at the Tokyo station (139.7500 $^o$E, 35.6917 $^o$N) for the period between 1 January 2006 and 31 December 2015 in JST.

A time window of 365 days was used as a database for predicting the corresponding weather variables for the subsequent 2 h, by sliding the time window every 1 min. Therefore, we predicted the precipitation during the time period between 1 January 30 2007 and 31 December 2015 in JST.

To obtain the control case, infinite dimensional delay coordinates (InDDeCs) were used to trace the 20 nearest neighbors within the time window (Hirata et al. 2015; Hirata & Aihara 2017a; 2017b). When conditioning data points of the past, the 20 nearest neighbors whose days were separated from the current day by multiples of $D$ days were traced using the same InDDeCs. Afterward, for each case, simple average points for $p$ minutes ahead of the 20 nearest neighbors were obtained. Subsequently, 35 the prediction errors for each prediction step were evaluated by calculating the mean absolute error. Finally, the mean absolute errors of the control case were compared with that of the conditioned past data. Detailed mathematical analyses are provided in Appendix. If there is $D$ day periodicity, it was assumed that the above predictions with $D$ as the greatest common divisor of the day differences between the current day and those of the used past data are better than similar predictions with $(D-1)$ or $(D+1)$ as the greatest common divisor.

## 3 Results


The results are presented in Fig. 1. For comparisons, we have prepared the persistence prediction, the mean prediction, and the auto-regressive (AR) model (Hamilton 1994). In the persistence prediction, we assumed that the most recent value continues for the next time steps. In the mean prediction, we took the average of precipitation for the whole the year of 2006, and made the average for the prediction for the next 9 years. In the AR model, we used 120-dimensional models to predict $p$ minutes 45 ahead directly so that the time scale for the inputs of the models matches the time scale for the predicted values.

First, we examined whether the precipitation in Tokyo has a one-week periodicity. The comparison of our predictions with $D = 6, 7$, or $8$ with that of the control case, shows improvement in predictions. However, there is no evidence that the prediction with $D = 7$ is better than that with $D = 6$ or $D = 8$. Therefore, the scenario in which precipitation has a one-week periodicity was excluded.

Similarly, we examined whether there is 14 days periodicity in precipitation in Tokyo. The results presented in Fig. 1 show that the time series predictions of precipitation can be improved by considering the past data with $D = 14$. In particular, when the past data was conditioned with either $D = 13$ or $D = 15$, the predictions deteriorated as compared to that with $D = 14$. The mean rank of the predictions between 1 and 120 min were obtained after arranging the above predictions for each $p$ min ahead between 1 and 120. The predictions with $D = 14$ achieved a mean rank of 1.7167, followed by predictions with $D = 15$ 55 (1.9083), $D = 13$ (2.4167), $D = 8$ (4.0250), $D = 7$ (5.0500), $D = 6$ (6.1833), the control case ($D = 1$) (7.3500), persistence prediction (7.5000), AR model (8.8500), and the mean prediction (10.0000).


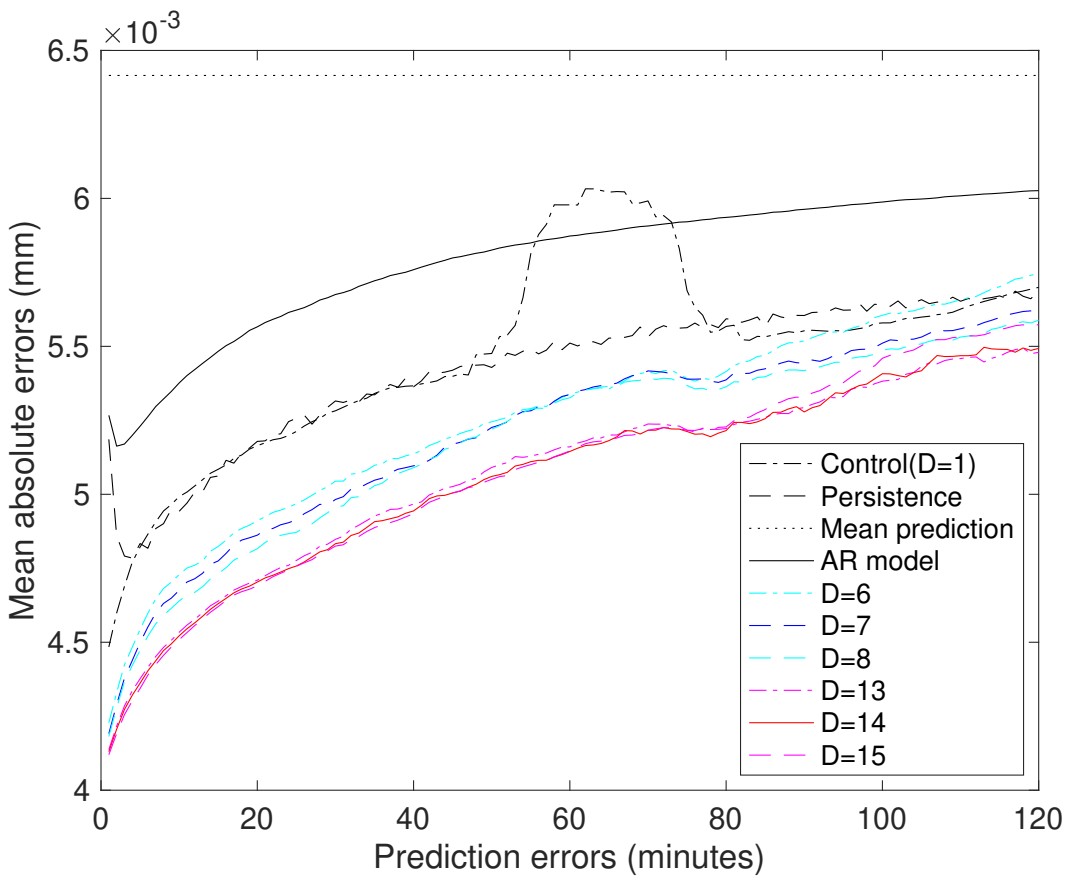

**Figure 1.** Mean absolute error of time series prediction for precipitation up to 2 h ahead.

## 4 Discussions

Temperature differences among days of the week in Japan have been reported by Fujibe (2010). In addition, Hirata and Aihara (2017b) reported that precipitation is concentrated downstream within the causal network among the weather variables, and is

affected by temperature. This is consistent with the results of this study.

Our results imply that days-of-the-week effects could take a more complicated form than simple weekly cycles of meteorological variables discussed in the existing literature. The results suggest that conditioning the past data every 2 weeks improves the precipitation prediction accuracy, implying that there is a period doubling bifurcation in the precipitation and that a week periodicity, if it exists, could be unstable. In addition, our approach could overcome the criticism of statistical tests related to

the previous results (Schults et al. 2007; Sanchez-Lorenzo et al. 2012). Moreover, our analysis was based on precipitation data for every minute, and thus the concern for poor data quality such as Viney & Bates (2004) could not be applied here. To further





investigate the implication of our findings, the effect of anthropogenic activities on the fortnight effects of precipitation should be examined carefully.

Overall, the results show that the time-series predictions of precipitation up to 2 h ahead can be improved by conditioning

the data of past days whose differences from the current day are multiples of 14 days. Our current results are limited to the observations at Tokyo. Hence, the practical significance of this methodology on weather forecasting in general should be comprehensively examined in the future.

*Code availability.* The codes used in this manuscript are available in the supplementary material in this manuscript.

**Appendix: Mathematical detail**

The mathematical details of the method applied are as follows. Suppose, the underlying dynamics are written by a function $f : M \to M$ as $x(t+1) = f(x(t))$, and a given time series is denoted by an observation function $g : M \to R$ as $s(t) = g(x(t))$, where $M$ represents an $m$-dimensional manifold. Infinite-dimensional delay coordinates (InDDeCs) (Hirata et al. 2015; Hirata & Aihara 2017a; 2017b) are an extension of delay coordinates (Takens 1981) for reconstructing the original state $x(t)$ from a series $\{s(t)\}$. Suppose, that an observational series $s(t)$ is given sequentially for $t > 0$. For $t \geq 0$, define $s(t) = 0$ for

convenience. Subsequently, InDDeCs $\boldsymbol{s}_\lambda(t)$ can be defined as $\boldsymbol{s}_\lambda(t) = (s(t), \lambda s(t-1), \lambda^2 s(t-2), \ldots)$. Here, $\lambda$ is a forgetting factor satisfying $0 < \lambda < 1$. Now, we set $\lambda = 0.95$ throughout the study. The $L_1$ distance in InDDeCs can be defined as $d(\boldsymbol{s}_\lambda(t_1), \boldsymbol{s}_\lambda(t_2)) = \sum_{\tau=0}^{\infty} \lambda^\tau |s(t_1 - \tau) - s(t_2 - \tau)|$. Based on this definition, the distance $d(\boldsymbol{s}_\lambda(t_1), \boldsymbol{s}_\lambda(t_2))$ can be obtained by the distance $d(\boldsymbol{s}_\lambda(t_1 - 1), \boldsymbol{s}_\lambda(t_2 - 1))$ by

$$d(\boldsymbol{s}_\lambda(t_1), \boldsymbol{s}_\lambda(t_2)) = |s(t_1) - s(t_2)| + \lambda d(\boldsymbol{s}_\lambda(t_1 - 1), \boldsymbol{s}_\lambda(t_2 - 1)). \tag{1}$$

Thus, alternatively, we used Eq. (1) to recursively define $d(s(t_1), s(t_2))$ for $t_1, t_2 > 0$ by setting the initial conditions, $d(\boldsymbol{s}_\lambda(t), \boldsymbol{s}_\lambda(1)) = d(\boldsymbol{s}_\lambda(1), \boldsymbol{s}_\lambda(t)) = |\max_{i \in [1, 60 \times 24 \times 365]} s(i)|$ for $t > 0$ so that we can avoid finding the $K$ nearest neighbors that are only close to $t = 0$.

The length of the time window was set as $l_w$, assuming that $t_c$ is the current time and the number of prediction steps is $p$. Subsequently, a series of distances $\{d(\boldsymbol{s}_\lambda(t_c), \boldsymbol{s}_\lambda(t)) | t = t_c - p, t_c - p - 1, \ldots, t_c - l_w\}$ were obtained for the current time $t_c$

and the $K$ smallest distances among them were traced. If, $I(t_c)$ be the time index for the $K$ smallest distances for time $t_c$; then, the usual time-series prediction for p steps ahead for time $t_c$ can be given by $\bar{s}(i + p | i) = \frac{1}{K} \sum_{i \in I(t_c)} s(i + p)$. This prediction becomes the control condition.

To consider the effects of possible $D$ day periodicity, we find $K$ smallest distances from the distances $\{d(\boldsymbol{s}_\lambda(t_c), \boldsymbol{s}_\lambda(t)) | t = t_c - p, t_c - p - 1, \ldots, t_c - l_w, \lceil \frac{t}{1440} \rceil - \lceil \frac{t_c}{1440} \rceil \equiv 0 \bmod D\}$. In particular, the time indices $t$ were selected such that the difference

between the corresponding day and the current day was a multiple of $D$ days. If, $J(t_c)$ be the time indices of the $K$ smallest distances among the redefined set of distances; then, the time-series prediction for $p$ steps ahead of time $t_c$ conditioned by a possible periodicity of $D$ days can be given by $\hat{s}(j + p | j) = \frac{1}{K} \sum_{j \in J(t_c)} s(j + p)$.





The mean absolute errors were used to evaluate the prediction errors. For the control case, we evaluated $E[|\bar{s}(i+p|i) - s(i+p)|]$ for $p$ steps before the prediction in the period between 1 January 2007 and 31 December 2015 in JST. To evaluate the

days-of-the-week effects, we evaluated $E[|\hat{s}(j+p|j) - s(j+p)|]$ for $p$ steps ahead, of prediction.

*Author contributions.*   YH and YY conceived the research. YH analyzed the datasets. YH and YY discussed the results of the study. YH drafted the manuscript. YH and YY edited the manuscript.

*Competing interests.*   The authors declare that there is no competing financial interests related to this manuscript.

*Acknowledgements.*   We appreciate the Japan Meteorological Agency for providing the datasets used in this study. This research was sup-

ported by JSPS KAKENHI Grant Number JP19H00815 for the project titled "Clarification of generating mechanisms of cloud-related severe phenomena over diverse surfaces by advanced methods and the improvement of their short-range forecasts".





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
