# Peer review of "Fortnight conditioning of historical data to improve short-term precipitation predictions"

_Nonlinear Processes in Geophysics, 2022_

## Author Comment (AC1)

Dear Dr. Dmitri Kondrashov

Thank you very much for providing your valuable comments to our manuscript.

Dr. Kondrashov's comment:

*I am simply not convinced by this paper, it is very short with one figure and is not up to the standards and depth expected for NPG.   Authors need to heavily revise and extend the manuscript to improve presentation and their arguments. Hopefully my comments below are helpful.*

Our response:

Thank you very much for your opinion. Our intention of the current manuscript is to provide a simple fact that time series forecasting of precipitation may be improved just by conditioning the past data. The Improvement of short-range forecasts of precipitation is very important mainly from the viewpoint of disaster prevention, the concrete methods for such improvements have not, however, been established yet. This paper proposed a better solution.   We do not mean to argue its underlying mechanism, as stated in the paper. We believe that the mechanism should be investigated later by the other experts by employing their expertise.

Dr. Kondrashov's comment:

*The authors argue that short-term (2hr ahead) time series prediction for precipitation at Tokyo station in 1-min sampling can be improved by using data two weeks in the past and some form of analogs method.   This is similar to looking for needle in a haystack and I find it very doubtful without additional analysis and presentation. First of all it would be helpful to show time series.*

Our response:

Please find the attached additional figure for the time series.

[Figure]

Additional Figure 1 | Time series of precipitation between 2006 and 2015

Dr. Kondrashov's comment:

*Secondly, are there any periodicities in the time series itself by using classical spectral analysis methods?*

Our response:

Please find the power spectrum of the precipitation as the second attached file. The power spectrum does not show a particular peak or some.

[Figure]

Additional Figure 2 | Power spectrum of the time series of precipitation

Dr. Kondrashov's comment:

*Finally, they should think on how to better present and illustrate their prediction method, perhaps using some toy model data, not simply as a short appendix.*

Our response:
Please find our third additional figure, which shows the results on the Rössler model forced by a periodic signal with period of "14 days". Then, we found that the time series prediction taking into account 14 day periodicity shows the better performance than those with 13 or 15 day periodicity, while the time series prediction with 7 day periodicity is competitive with that of 14 day periodicity. I hope that this toy example help you and the other readers understand our findings.

[Figure]

Additional Figure 3 | Prediction errors for the Rössler model by taking into account the various potential periodicities D.

---

## Author Comment (AC2)

Dear Referee #1,

Thank you very much for reading our manuscript carefully and providing us the insightful comments.

Referee's comment:

> *Review of the manuscript 'Fortnight conditioning of historical data to improve short-term precipitation predictions' by Yoshito Hirata and Yoshinori Yamada*

> *The present manuscript tries to attribute short-range precipitation predictability in the large Tokyo megalopolis to the indirect effect of aerosols produced by anthropogenic activities, through their influence on the production of precipitation nuclei and optic effects.*

Our response:

Thank you very much for your comments here. Our intention here is to show the fact that short-term time series prediction of precipitation may be improved by conditioning past data periodically. However, we do not mean to argue the underlying mechanisms of why this periodicity is generated.

Referee's comment:

> *The manuscript is very short not giving enough details for the appropriate reproducibility of the results. Moreover, the methodology, and the arguments in the discussion are very dubious and even not falsifiable, which is fundamental requirement in any scientific theory. Moreover, there are severe methodological shortcomings, described below. Giving those reasons, the manuscript is judged not reaching enough standards to be published in NPG.*

Our response:

Thank you very much for sharing your views. However, we do not agree with your views because (i) our message is quite simple: we can improve short-term time series prediction of precipitation by just conditioning past data periodically. This message itself should be practically important enough to be published so that we may be able to prevent causalities due to the heavy rains; (ii) we have described our method concisely in the Appendix. In addition, we have attached our codes so that readers can reproduce what we have done; (iii) In the reply to Dr. Kondrashov, we supplied the results of a toy example forced periodically. This example demonstrates that by conditioning past data periodically, we can take into account the periodicity of the underlying dynamics in our time series prediction.

Referee's comment:

*The present study should be preceded by experiments with a toy minimal model, reproducing convection and precipitation mechanisms triggered by aerosol nucleation. Then, predictability experiments should be run by imposing some weekly periodicity to aerosol emissions to simulate the periodic anthropogenic forcing and seek whether any phase synchronizing is observed in precipitation. The predictability study described in the manuscript, obtained with timeseries only is far unsatisfactory due to the existence of a vast number of noncontrolled factors, beyond aerosols. It is thus very difficult to produce a convincing quantifiable attribution of the very-short term precipitation predictability to the aerosol's forcing.*

Our response:

Thank you very much for your comments here. In the reply to Dr. Kondrashov, we have provided a toy example. Thus, we have shown that by the proposed approach, we could consider the underlying periodicity in short-term time series prediction. As mentioned above, we do not intend to discuss the underlying paths on why this periodicity occurs because (i) our finding itself has the applicational values and (ii) we do not have the expertise to discuss the underlying mechanisms. By establishing the fact

that time series prediction can be improved by conditioning past data periodically, then the underlying mechanisms should be further investigated by the other researchers who know better the underlying physics of precipitation.

Referee's comment:

*The applied methodology is dubious and impacted by severe pitfalls such as:*

- *The method of analogues is too little described; for instance, the analogs metric is not clear. Is it based on precipitation only? If yes, the analog's distance is too strict.*

Our response:

Thank you very much for sharing your view. Yes, our time series prediction is only based on past data of precipitation. In our prediction procedure, we do not have to run a big meteorological model or some so that we can issue time series predictions in 1 minute resolution up to 2 hours ahead. We will consider combining the other weather variables as a future research topic. Thanks.

Referee's comment:

- *It is not clear if analogs are sought in an independent period of the validation period.*

Our response:

Predictions were made by using the information up to the certain time and thus independent of the future values of precipitation.

Referee's comment:

- *The details of the AR model are not described. Other benchmark stochastic models should be tested.*

Our response:

We are sorry that we have not described the mathematical detail of the AR model, although we had explained by words in lines 44-45. In the AR model, we fit the following model by the least square fitting:

$$\hat{s}(t + p) \sim a_{p,-1} + \sum_{d=0}^{119} a_{p,d} s(t - d).$$

We used the dataset of 2006 to find the parameters $a_{p,d}$ and evaluated the prediction errors on the dataset between 2007 and 2015.

We presume that the AR model should be one of our benchmark stochastic models. We also have included the persistence model as well as the mean prediction model as the other benchmarks. Therefore, if you think that we should include the other benchmark stochastic models, could you raise an example of such methods or some to be included so that we can evaluate more concretely whether we should consider the other benchmarks or not.

Referee's comment:

- *By forecast rank, authors mean error, so authors should precise that.*

Our response:

We ranked the mean absolute error for each method for each prediction steps. Then, we took the mean of the ranks over the prediction steps. We hope that our meaning is clearer.

Reviewer's comment:

*The unique figure presented is not fully discussed. There are results which are not understandable neither discussed such as: the bump in rank around the forecast delay 60-70 minutes for D=1; the reason why the predictability is larger when analogous are sought with D=14 than D=7. Authors present a very speculative unproven reason for that: 'there is a period doubling bifurcation in the precipitation and that a week periodicity, if it exists, could be unstable'.*

Our response:

Thank you very much for your comments here. For D=1, we have observed that 65 minutes ahead prediction forecasted more rains than 50 or 80 minutes ahead prediction when it did not rain actually as shown in Additional Figure 4. In addition, we decided to withdraw our statement related to the above period doubling bifurcation. Based on the toy example of the periodically forced Rössler model, we can say that we could do better in the medium-term time series prediction when we turned the periodicity of the conditioning to the right period. As far as we tested the period doubling could not be observed in the toy model. We deeply appreciate your critical comments, by which we could remove this wrong statement.

[Figure]

Additional Figure 4 | Histogram for predicted precipitation for 50, 65, and 80 minutes ahead predictions when we set D=1.

---

## Author Comment (AC3)

Dear Referee #2

Thank you very much for reading our manuscript critically.

Reviewer's comment:

> *This paper illustrates how conditioning a nowcast precipitation prediction on the calendar day improves a precipitation score.*
>
> *The poor English syntax and grammar make the manuscript very difficult to follow. The lack of clear and detailed explanations of what is done make the manuscript impossible to assess. For example, the annex is not called in the main text and it does not clarify anything on the procedure that is used by the authors.*
>
> *From what I see, I do not see how the paper is relevant in NPG, as I do not see a real conceptual innovation (the only innovation appeared in a paper already published by the authors).*

Our response:

Thank you very much for sharing your view. But, we believe that our finding of the possibility that only conditioning past data improves a time series prediction is novel conceptually because we do not need additional measurements and thus almost free. Our mathematical details are given in the Appendix. Here, we had cited the Appendix in Section 2 in the original submission. Moreover, we have included our codes as the supplementary material. Thus, we believe that the readers can reproduce what we have done.

Reviewer's comment:

Therefore, my appraisal of the paper is based on a guess of what was done to obtain the results.

Major points

The authors use time series with a time increment of one minute. Therefore, not only there is a seasonal cycle, but the time series also contain a diurnal cycle. If there is any cycle in the data, a Fourier transform should be able to detect it. The results reported in Figure 1 do not suggest any type of periodicity.

Our response:

Here, we have two things to state: (i) our intention is to predict precipitation in 1 minute resolution up to 2 hours. Therefore, we need to predict a diurnal cycle if it exists. (ii) We do not state in the conclusion that there are two week cycles in precipitation. No periodicity is considered in the proposed model.   We rather want to state that there are two week dependence. Even if there is not a two week cycle, there could be two week dependence. By exploiting such two week dependence, we could improve time series forecasts as demonstrated in Figure 1. Thus, if we are allowed to revise the manuscript, we will rephrase the word "cycle" in Section 1 to "dependence" so that we can contrast "dependence" with "cycle".

Reviewer's comment:

The methods section does not state how precipitation is predicted (e.g. what model?). Even the AR prediction is not clear. How are the authors certain that they do not over fit the data?

Our response:

We described the mathematical details of our method in Appendix. If we summarize our prediction by words, our prediction is of 20 nearest neighbor prediction based on infinitely dimensional delay coordinates. AR model is described in the reply to Referee #1. Since we use the dataset of 2006, which have about 525600 points, we do not overfit when we fit only 121 parameters.

Reviewer's comment:

*When they use the term "improve" (e.g. in the title), they should state with respect to what? The improvement over operational nowcasting from meteorological institutions should be demonstrated.*

Our response:

First, we are predicting precipitation in 1 minute resolution, which operational nowcasting is not doing as far as we have read Ravuri et al., Nature 597. 672-677 (2021). Second, mostly common methods for the current nowcasting of precipitation rely on the extrapolation of radar echoes to forecast areal rainfall for the temporal resolution of around several to 10 minutes, while we only use precipitation measurements at a point.   The current one gives us information of the "areal pattern" and "degree of rainfall intensity" of future precipitation fields.   One of our originality lies in the point-to-point forecast of precipitation at a very high temporal resolution of one minute.   Among methods available to forecast a high temporal resolution, we found that the proposed method has a superiority or "improvement" relative to other methods.   There seem no nowcasting methods that permit point-to-point forecast with such high temporal resolution.   This method may be easily extended to the areal precipitation forecast.   Third, we intend to show the fact that simply conditioning the past data can improve time series prediction compared with the case without conditioning. We reached the similar results with the same dataset of Tokyo using 20-dimensional usual delay coordinates (Additional Figures 5-9), although there is some year-to-year variability. Thus, we can apply our findings to the other forecasting methods to improve such forecasting at least at Tokyo.

[Figure]

Additional Figure 5 | Precipitation prediction of year 2007 by dataset of 2006 with 20-dimensional delay coordinates. We took the simple average of 20 nearest neighbors in the dataset of 2006 to predict up to 120 minutes ahead in year 2007. Therefore, this result is one of cross validations.

[Figure]

Additional Figure 6 | Precipitation prediction of year 2008 by dataset of 2007 with 20-dimensional delay coordinates. Please see the caption of Additional Figure 5 to interpret the results.

[Figure]

Additional Figure 7 | Precipitation prediction of year 2009 by dataset of 2008 with 20-dimensional delay coordinates. Please see the caption of Additional Figure 5 to interpret the results.

[Figure]

Additional Figure 8 | Precipitation prediction of year 2010 by dataset of 2009 with 20-dimensional delay coordinates. Please see the caption of Additional Figure 5 to interpret the results.

[Figure]

Additional Figure 9 | Overall summarized results shown for Additional Figures 5-8. These results indicate very well the higher performance of the proposed method by the physically-meaningful magnitudes of errors.

Reviewer's comment:

I feel that the reported result (better 2h forecast when taking D=14 day prior information) is only valid for the statistical scheme alluded to by the authors. Nothing proves or even suggests that this would hold in a "regular" nowcasting meteorological forecast.

Our response:

As we discussed above, the establishing fact that precipitation at Tokyo has two-week dependence can be applied to the other methods because what we do is just conditioning the past data. In addition, we have shown that

we could obtain similar results even if we use 20-dimensional delay coordinates as shown above in Additional Figures 5-9.    In addition, if a-point-to-point precipitation forecasts will be available at many stations ensemble in an target area, the current nowcasting method would be replaced by the proposed method because most of radar-echo-based extrapolations do provide information on areal extent and the degree of intensity of precipitation.

Reviewer's comment:

*The main result of the paper is based on Figure 1. But this figure does not prove anything, in particular for D=14. The authors have not tried other values of D, in particular larger values. The seasonal dependence is not discussed or even assessed. Why is there a "bump" for D=1? Precipitation differences of 0.006 mm (maximum value of the vertical axis in Figure 1) are not measurable by meteorological instruments. Therefore, the apparent minimum for D=14 cannot be measured in practice. This minimum of mean absolute error might not even be statistically significant (and it is obviously not physically relevant).*

Our response:

As we replied to Referee #1, we have observed for D=1 that 65 minutes ahead prediction forecasted more rains than 50 or 80 minutes ahead prediction when it did not rain actually as shown in Additional Figure 4 in the reply to Referee #1. Because, we have evaluated our predictions about 4733280 time points and we have 0mm for the most times, our prediction errors become as small as 0.006mm. Due to these two reasons, we obtained this small but physically relevant number. We also showed the seasonal variation in Additional Figure 10. Our findings have the stronger tendency for Autumn and Winter. The tendency we found is statistically significant. For example, in the results shown in Additional Figures 5-8, D=14 shows the smallest errors among the 7 tested predictions for 204 out of 480 prediction steps. If we apply the binomial distribution, assuming that each prediction is independent, and D=14 becomes the best prediction with the probability of 1/7, then p-value becomes smaller than $1.0 \times 10^{-16}$.

[Figure]

Additional Figure 10 | Seasonal dependence of our findings. In these panels, we used the same colors and same styles for the lines as Figure 1.

Reviewer's comment:

The right way to assess forecast schemes is to use cross validation procedures, i.e. at least by considering a training period and a separate validation period. Therefore, it is not even clear that the reported result is actually true.

Our comment:

Because our dataset contains 5258880 time points, we needed to use a computational efficient method, which is Hirata et al. (2015). In this method, we cannot clearly split a training part and validation part of dataset because a time series is processed along the time axis. But, when

we predict values, we only use the time points up to then. Thus, we clearly follow the causation of the given time series. Moreover, in the above results, we use the 20-dimensinal delay coordinates to clearly distinguish training datasets and validation datasets. Then, we obtain the similar results shown in Additional Figures 5-9. Although there is some year-to-year variability, overall the prediction only using multiples of 14 past days tends to have the highest prediction accuracy. Thus, we believe that our results can hold even if we change the method of prediction.

Reviewer's comment:

Specific comments

Abstract: is precipitation dependence a weather variable? (or what is weather variable, and how do the authors define "precipitation dependence"?).

Our response:

We apologize of this confusing expression. Here "The effects of changes in weather variables" include "precipitation dependence on the days-of-the-week".

Reviewer's comment:

*l. 15: Why and how the uni-modal relation (whatever that means) of aerosols and convective energy (why convective energy) is connected to the scattering and absorption of solar radiation?*

Our response:

Because aerosol optical thickness gradually increases and aerosol transmission gradually decreases while the amount of aerosols increases, convection energy results in a uni-modal function in terms of the aerosol concentration, according to Fig. 4 of Rosenfeld et al. (2008).

Reviewer's comment:

*I think that the authors miss the main point of predicting precipitation, as they treat zero values in the same way as non zero values.*

Our response:

Here, we treat zeros and non-zero values of precipitation in the same way. The mathematical detail of the used method is described in Appendix.

Reviewer's comment:

*The methods section is inappropriately unclear, especially for a journal like NPG. The first paragraph of section 3 should be in the methods section. The AR model is not defined properly. An order of 120 sounds like overfitting. Precipitation is not Gaussian, especially at minute time scales. An AR model is an obvious bad choice.*

Our response:

As we mentioned in Section 2, the mathematical detail is described in Appendix. If you think the mathematical detail should be described in the main text, we will move the contents of Appendix to Section 2. As we discussed above, in our setting, we do not overfit the AR model with 121 parameters because there are about 525600 time points. The AR model is a model the Editor suggested during the initial decision. If you kindly raise an alternative that fits our current setting, we will consider it in the revision.

Reviewer's comment:

*Why don't the authors consider the hour of the day when they condition the forecast? They might avoid an aliasing phenomenon that could explain a fortnight conditioning.*

Our response:

If we consider the hour of the day, we might be able to improve our results more. But, this is out of the current scope. There is some human behavior that occurs bi-weekly around Tokyo such as garbage collections of nonburnable.   Therefore, we could not deny our finding as an "aliasing phenomenon".   The increase in the aerosol concentration may not, however, be sufficient to account for the two-week dependence of precipitation because not all of aerosol particles do not act as condensation nuclei.   Hence, we would like to first establish a solid fact that the detailed precipitation at Tokyo has two-week dependence. Then, we would like to examine its mechanism later.

Reviewer's comment:

*The first paragraph of section 4 is incomprehensible, and is not related to analyses of the paper.*

Our response:

It is our discussion behind our finding. Temperature at Tokyo has week dependence (Fujibe, 2010). Precipitation is at the downstream in a network of weather variables (Hirata & Aihara, 2017). Especially, temperature influences the precipitation. Thus, it is natural to consider that precipitation also has week dependence or more complicated dependence.

Reviewer's comment:

*The Appendix section is not really informative on what is done in the forecast.*

Our response:

This is our mathematical detail of what we have done. If it is appropriate to move this Appendix to the main text, we will do so during the revision phase. Furthermore, we have attached our codes as the supplementary material. Thus, we could ensure that our work can be reproduced by readers.

Our response:

> *Conclusion*
>
> *I cannot recommend the publication of this manuscript in NPG.*

Our response:

We appreciate you again for reading our manuscript critically. But, as we have discussed above, we can remove most of your concerns, strengthening our findings. If there are some parts that are difficult to understand, we will revise these parts in the revision.